# Probabilistic graphical modelling of early childhood caries development

**Alessandro Ugolini**[1]*, **Francesco Porro**[2], **Federico Carli**[2], **Paola Agostino**[1], **Armando Silvestrini-Biavati**[1], **Eva Riccomagno**[2]

1 Department of Surgical and Integrated Diagnostic Sciences, University of Genova, Genoa, Italy,
2 Department of Mathematics, University of Genova, Genoa, Italy

* alessandro.ugolini@unige.it

**Data Availability Statement:** The complete dataset is published online at: Ugolini, Alessandro (2022), "Early childhood caries development", Mendeley

## Abstract

In order to develop a statistical model for complex interactions among factors affecting early childhood caries development (ECC), 234 children from the "XXXX oral growth longitudinal study" were analysed at ages 3, 4 and 5. A questionnaire for children's parents (clinical history, nutritional and oral hygiene habits) and clinical oral examinations were recorded each year. The probabilistic dependence structure on the most significant factors was modelled with an Undirected Graphical Model (UGM or Markov random fields) which provides a probabilistic reasoning approach for the establishment of multi-way associations. The best fitting UGM was estimated through the maximum likelihood principle and two-way factor associations were verified through Fisher's exact statistical hypothesis tests for count data. The effect of sugar intake on caries incidence is mediated by oral hygiene; for caries incidence, oral hygiene quality is more relevant than toothbrushing time; the effect of pacifier on caries incidence is statistically significant only when considered in combination with breastfeeding time. Among behavioural ECC risk factors, the quality of oral hygiene, and not the toothbrushing frequency, is a primary factor that modulates the sugar intake in his primary role of the ECC developer. School-based oral health programs for ECC prevention should be improved with supervised toothbrushing program. UGM analysis could improve the school-based oral health programs with more effective and efficient prevention strategies based on the hierarchical interactions among the ECC risk factors. Oral hygiene plays a pivotal role in early childhood caries and can modulate positively or negatively their development. Supervised toothbrushing is a crucial intervention to be included in the daily educational and clinical practice and in the school-based oral health promotion programs.

**Trial registration: Clinical trial registration number**: NCT02798809.

## 1. Introduction

"Early Childhood Caries (ECC) is defined as the presence of one or more decayed (non-cavitated or cavitated lesions), missing or filled (due to caries) surfaces, in any primary tooth of a child under six years of age" [1]. ECC is the most common chronic infectious disease of childhood with the prevalence up to 65% and is becoming a serious public health problem in both

Data, V1, doi: 10.17632/7rsxyfsnd5.1 Direct link:
https://data.mendeley.com/datasets/7rsxyfsnd5.

**Funding:** The authors received no specific funding
for this work.

**Competing interests:** The authors have declared
that no competing interests exist.

developing and industrialized countries. ECC typically begins early in life at around 3 years old and progresses rapidly in those who are at high risk, and often goes untreated. Its consequences can affect the immediate and long-term quality of life of the child and family and can have significant social and economic consequences. ECC has been at epidemic proportions in the developing countries [2].

Dental caries results from complex interactions among acid-producing members of the biofilm, fermentable carbohydrates, and many patient factors, including enamel weakness and saliva critical pH. Recently, a few studies pointed out the role of the genetic factor in caries. But basically, caries is mainly modulated by behavioural and environmental risk factors, such as diet, fluoride exposure and dental hygiene and currently is considered a diet-mediated disease [2–4].

Diet habits, especially sugar intake and oral hygiene, are the most important key factors in the caries prevention. Several systematic reviews [5–7] emphasized their role in the caries development and most of the school or community-based interventions for caries prevention focused their action on a behavioural approach to establish and maintain effective oral health routines and dietary recommendation [8–11].

In the present study Undirected Graphical Models (UGMs, also known as Markov random fields) are applied to assess the complex interactions among factors affecting ECC. UGMs offer a probabilistic reasoning approach based on graph theory for the establishment of conditional independence relationships among the study factors, thus expressing the interaction of a complex system in probabilistic terms. Their graphical structure provides a visualization of those independences. The UGM estimated from a dataset can be useful to confirm known independence relationships, to validate the dataset and mainly to identify unexpected relationships [12]. UGMs are applicable in many fields, can be estimated also in presence of latent/unobserved variables or missing values, supply a compact representation of conditional independences and correlations among study factors, and model updating is a local computation involving only variables within cliques. (Cliques are subsets of vertices such that every two distinct vertices are connected by an edge). Furthermore, they do not only allow the encoding of probability distributions but also provide a very clear interface to interpret the model and to perform predictions. They do not require necessarily the definition of response variables, as often done while analysing contingency tables. The main consequence of the fact that there is no need to distinguish a-priori response variables and covariates, is that all interactions among all variables are captured in a single model equation (of the joint probability distribution) including interactions among response variables themselves. This can lead to the identification of unexpected relationships and allows the exclusion (or inclusion) of a-priori unexpected conditional independences. More technically, UGMs are statistical models for estimating the joint probability distribution of a random vector. Many linear regression models are UGMs. For categorical variables, as those in the present study, a UGM corresponds to a factorization of the joint probability density function for the random vector $X$ (modelling all the study variables). The response variable to be modelled is the probability of any possible value taken by $X$, namely any combination of the study variables, estimated under the UGM. Importantly, each factor of the factorization relates to a subset of the components $X$, each subset can be modelled by a UGM separately from the other subsets in the factorization and it can be estimated by using only a marginal table of the data (thus reducing the dimensionality of the multivariate study) and then multiplied together to return the full joint probability [13,14].

In this study the UGM approach is adopted on a longitudinal dataset, with the purpose of validating existing knowledge on the interaction of the risk factors associated with ECC, of investigating the existence of further interactions and of assessing the overall relationship among ECC risk factors. An earlier application of the log-linear models in longitudinal

pediatric dentistry [15] studies the effect of dental health education on children's use of dental health care services. The graphical representation of a UGM provides also a visual tool which is helpful in communicating and reasoning on the results of the analysis.

## 2. Material and methods

### 2.1 Study design

The present research starts from the dataset obtained from the "XXXX oral growth longitudinal study (GeOrGS)", an observational longitudinal study approved by the Ethical Committee of University Hospital "XXXXXXX", XXXXX, XXXXX (protocol number 4/2012) and by the Local Health Centre XXXXXXXXX (trial registration: ClinicalTrials.gov Identifier NCT02798809, https://clinicaltrials.gov/ct2/show/study/NCT02798809). All procedures performed in in the present study were in accordance with the ethical standards of the institutional research committee and with the 1964 Helsinki declaration and its later amendments or comparable ethical standards. Informed consent was obtained from all individual participants included in the study.

It was conducted in preschools of XXXXX (27,000 inhabitants), a town nearby XXXXXX. All children of XXXXX District born in 2008 and 2009 were enrolled in the study. At each visit two types of data were collected: data related to the clinical examination and data related to children history and habits. The latter were collected with a questionnaire that was filled out by the parents before each clinical examination, and included questions about children's clinical history, nutritional habits and oral hygiene and the consensus to the clinical examination. The first clinical examination was performed at 3 years old, then repeated at 4 years old and at 5 years old. Recruitment is currently underway to increase follow-up time to 10 years old.

### 2.2 Participants

*Inclusion criteria* were all children born in 2008 and 2009, attending one of the 10 kindergartens in Chiavari, for whom written parental consent to visit, to study participation and to personal data processing was given, child's presence at school on the day of the clinical examination.

*Exclusion criteria* were incorrect or incomplete questionnaire's filling by parents; no compliance (visit rejection by the child), alterations in the number, size and shape of teeth; syndromes or systemic problems affecting craniofacial growth; cleft palate; undergoing orthodontic treatment.

The initial sample consisted in 388 children. 63 parents denied consent to visit and the number of children visited and enrolled in the study was 325. One child with Down's syndrome was enrolled and excluded only after the clinical examination to support his desire to take part to his fellow's activities, 18 children had incomplete questionnaires, 12 children had dental alterations in size/shape or number, and 11 were absent for 2 of the 3 visits and 6 had previous or ongoing orthodontic treatment. The final sample consisted of 277 children.

### 2.3 Clinical examination and data collection

This report is part of a wide longitudinal study on children's oral health, data collection included a survey on feeding, sucking and hygiene habits. Two WHO (World Health Organization) -calibrated examiners carried out the clinical examinations from 2011 to 2014. They were blinded to the information collected from the parental questionnaires. Dental examination was performed in the classroom environment using disposable gloves and masks in compliance with the infection control protocol, sterilized mouth mirrors and probes. Data were

collected according to WHO criteria [16]. Oral hygiene status was defined as adequate (1) or not adequate (2) after clinical examination, based on the absence or the presence of visible plaque on tooth surfaces (plaque index) [17]. Radiographic examination was not performed. Two senior examiners (P. A. and A. U.) with previous experience in epidemiological surveys participated in the study. The calibration process was performed at WHO Centre of Collaboration for Epidemiology and Community Dentistry (University XXXXXX). It began with a theoretical phase according to WHO criteria in order to verify examiners' knowledge about epidemiological diagnosis. After that, clinical training sessions were completed with a discussion among the examiners and the WHO trainer, with regard to clinical diagnosis and criteria used, recording the other errors. Following the clinical training exercises, the examiners undertook two calibration exercises for the WHO diagnostic criteria.

## 2.4 Error of method

Two WHO-calibrated examiners, blind to the parental questionnaire information, conducted the children's oral examination. At each of the three time points of the study an examiner calibration was carried out and during each clinical data collection 15 children (randomly selected) were visited twice by both examiners in order to verify the inter- and intra-examiner error for detection of decayed, missing and filled teeth and for oral hygiene status assessment. Kappa values for the inter- and intra-observer agreement for these variables were respectively calculated: Decayed, missing and filled teeth, 0.951 (95% CI: 0.922–0.989) and 0.934 (95% CI: 0.901–0.974); Oral hygiene status, 0.942 (95% CI: 0.914–0.978) and 0.929 (95% CI: 0.883–0.968). Overall, the method error was considered negligible.

## 3. Data analysis

### 3.1 Data preparation

The original dataset has 109 variables and 277 observations/children. The variables can be divided into three macro groups: 4 unit identification variables, 18 general questionnaire variables and 29 variables measured at three different time points, namely at age 3, 4 and 5 (Supplementary material 1, S1 Table in S1 File). The only drop out from the study was excluded as well as three observations for which important factors were not observed. Of the remaining 273 observations, 39 correspond to children from 12 different ethnicities (among them Albania, Romania, China, Morocco, Ecuador), while the other 234 are Italian-born children with Italian-born parents. The analyses carried out on the datasets with 273 and 234 observations identify the same risk factors for ECC, as outlined in the Section Results (see also Section 3.3). The main text, S2 Table in S1 File refers to the sample of size 234 while Supplementary Materials 2 reports results for the larger sample. The eight variables are constructed from the original variables in S1 Table in S1 File based on preliminary explorative data and association analyses [18]. Child age and the longitudinal nature of the dataset are incorporated into the definition of the eight variables. This construction is summarized below and sample descriptive statistics about the eight variables are reported in Table 1 and Fig 1.

1. *Breastfeeding type* and *Frequency of toothbrushing* are the original measured variables and give the type of feeding and the number of times per day the child brushes his/her teeth, respectively. *Breastfeeding type* takes value 0 if the child was not breast fed, 1 if the child was exclusively breast fed and 2 if the child was both breast and bottled fed. *Frequency of toothbrushing* is 1 if usually the child brushes his/her teeth once per day, 2 if twice or 3 if more than twice a day.

**Table 1. Relative/Percentage univariate distributions of the eight selected variables used for the UG modelling.**

| Caries incidence | % count | Oral hygiene status | % count |
|---|---|---|---|
| 0 (no variation) | 80.34 (188) | 1 (adequate) | 84.19 (197) |
| 1 (variation) | 19.66 (46) | 2 (not adequate) | 15.81 (37) |
| Breastfeeding type | | Frequency of toothbrushing | |
| 0 (not breastfed) | 18.38 (43) | 1 (once a day) | 27.35 (64) |
| 1 (exclusive breastfed) | 58.12 (136) | 2 (twice a day) | 60.68 (142) |
| 2 (breast and bottled fed) | 23.50 (55) | 3 (more than twice a day) | 11.97 (28) |
| Consumption of sugared beverages | | Consumption of vegetables/fruits | |
| 1 (daily) | 29.91 (70) | 1 (daily) | 41.45 (97) |
| 2 (weekly) | 56.84 (133) | 2 (weekly) | 53.42 (125) |
| 3 (occasionally) | 13.25 (31) | 3 (occasionally) | 5.13 (12) |
| **Breastfeeding time** | | **Use of Pacifier** | |
| 0 (0 months) | 18.38 (43) | 0 (0 months) | 31.62 (74) |
| 1 (1–6 months) | 30.34 (71) | 1 (1–36 months) | 29.06 (68) |
| 2 (7–12 months) | 36.32 (85) | 2 (36–48 months) | 24.79 (58) |
| 3 (> 12 months) | 14.96 (35) | 3 (> 48 months) | 14.53 (34) |

2. *Breastfeeding time* indicates for how many months the child was breastfed: 0 stands for zero months, 1 for breastfeeding up to six months, 2 for 7–12 months of breastfeeding, and 3 for more than 12 months. Also *Use of Pacifier* was constructed by categorization of an original variable and takes four values: 0 for never taken pacifier, 1 for up to three years, 2 between three and four years, and 3 for more than four years.

3. *Caries incidence* records whether the number of carious lesions and fillings varies from age 3 to age 5: it takes value zero if there is no zero increment or negative increment and one if there is a positive increment. It is constructed from the original variables as the difference between the sum of the number of carious lesions and fillings at age 5 and the sum of the number of carious lesions and fillings at age 3. At age 3, only 15 children had at least a carious lesion. For 46 subjects, the number of carious lesions and fillings increased. Only for three children the increment was negative.

4. *Consumption of sugared beverages* and *Consumption of vegetables/fruits* summarize the consumption of carbonated and sugary drinks and vegetables, respectively. They are a combination of original measured variables and take three values for daily (coded as 1), weekly (coded as 2) and occasional consumption (coded as 3). Their construction takes into account the original variables related to drinking and eating habits over the three years of the study: we collapse the measured values for drinking and eating habits at age 3, 4 and 5, thus a sort of average of these values is carried out. This was possible thanks to the non-significant intra-subject variability of these variables

5. *Oral hygiene status* is a measured binary variable with value 1 for adequate or 2 for not-adequate hygiene after a clinical examination, performed by two senior examiners as described in Section 2.3.

## 3.2 Model estimation

UGMs are used to detect multi-way associations which are expressed through marginal and conditional independences. The UGM estimated in our study is displayed in Fig 2. Its mathematical interpretation is given in this section, while its clinical interpretation is given in the Results section. The eight study variables are modelled by a random vector $X$ which takes

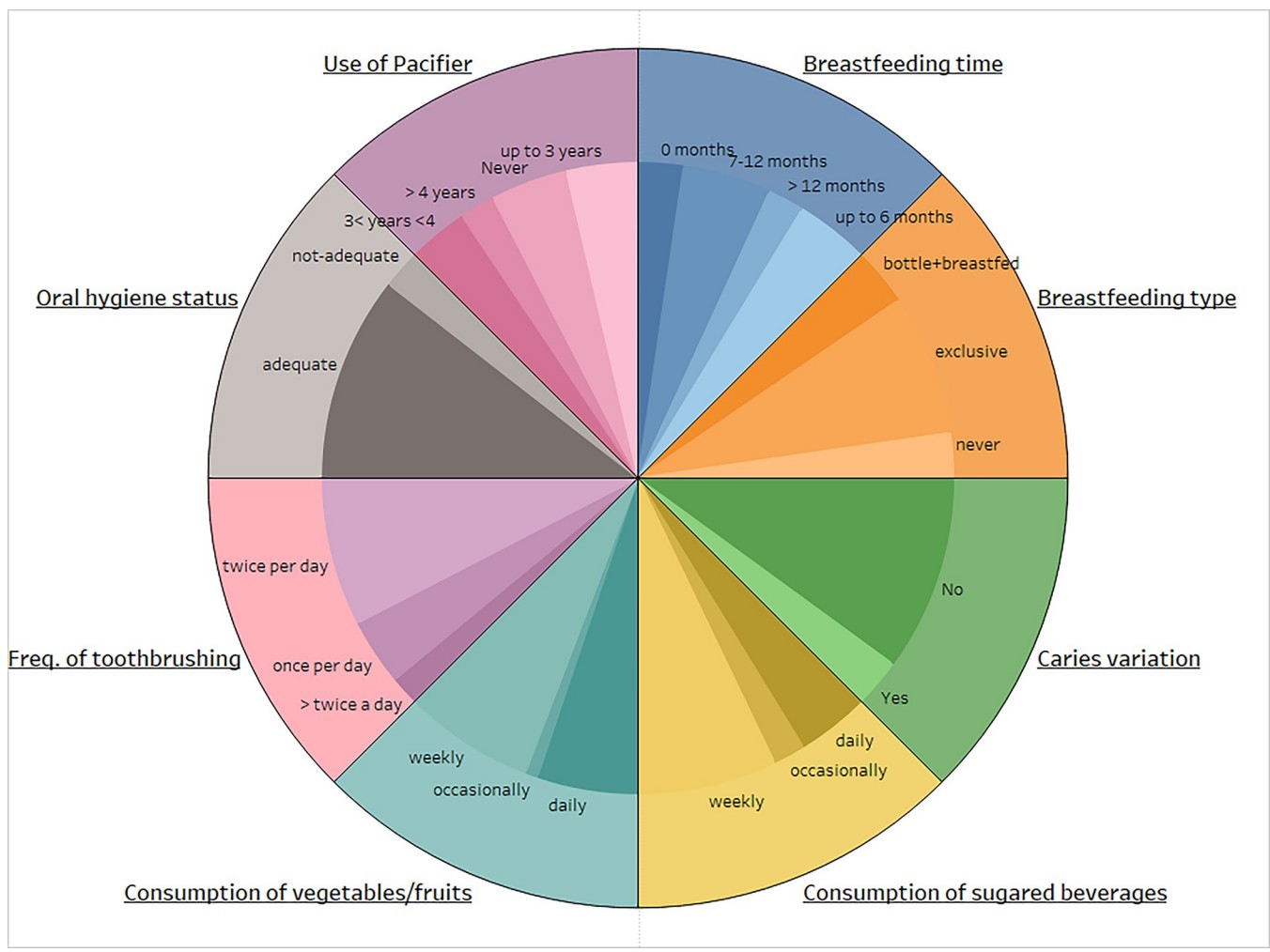

**Fig 1. Multilevel nested pie chart showing the distributions of the variables included in the UGM.**

values in a finite but large set, namely with $4^2 3^4 2^2$ (5184) possible combinations of values. Any such combination is indicated with $x$. The UGM in Fig 2 gives the following factorization of the joint probability distribution of $X$:

$$\mathcal{P}(X = x) = \mathcal{P}(X_V = x_V) * \mathcal{P}(X_B = x_B) * \mathcal{P}(X_R = x_R), \tag{1}$$

where $\mathcal{P}(X = x)$ is the probability that the eight study variables take the value $x$. Under the UGM the eight variables are decomposed into three macro factors as in Eq (1). The factor $\mathcal{P}(X_V = x_V)$ is the probability that the variable *Consumption of vegetables/fruits* takes value $x_V$ where $x_V$ can be 1, 2 or 3 (Table 1). Similarly, $\mathcal{P}(X_B = x_B)$ is the estimated probability of the variable *Frequency of toothbrushing* ($X_B$). The remaining six factors are collected in $X_R$.

In a framework with only categorical variables and being the probability of each instance of $X$ strictly positive, an UGM corresponds to a log-linear model [12,13,21]. The parameters of the log-linear model for the six connected variables in Fig 2 correspond to the intercept, one for each node (that is, each variable or main effect), one for each edge (that is, each two-way interaction). No three-way or higher level of interaction among variables has to be included in the log-linear model because in the estimated UGM's graph in Fig 2 there are no triangles nor

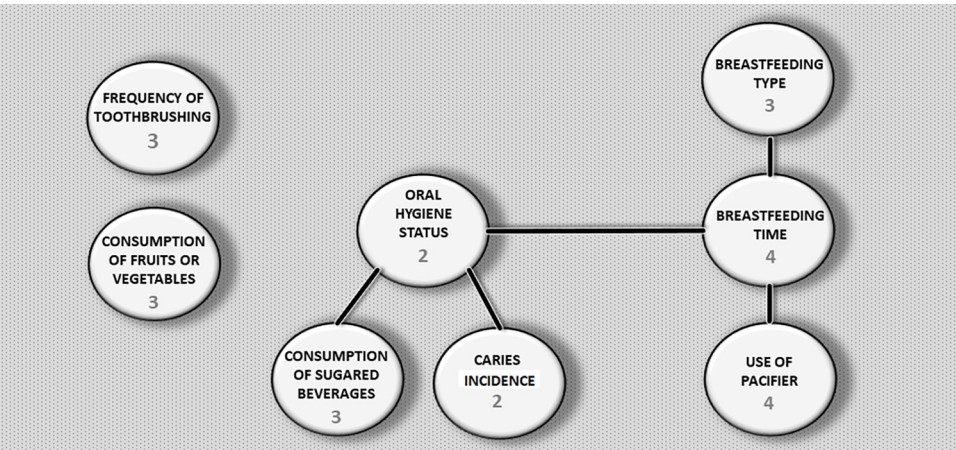

**Fig 2. Estimated UGM on the eight selected variables.** The number of levels of each variable is written in the variable node. A connection through an edge between a pair of nodes indicates a dependence relation among the corresponding pair of variables. A general conclusion is that two not directly connected nodes indicate a conditional independence between the two corresponding variables given all the other variables. In particular, a more detailed conclusion can be made: two not directly connected nodes that are connected through a path are independent conditionally only on the set of variables from which all paths connect the two nodes pass [Højsgaard et al., 2012]. For instance, caries incidence is independent from consumption of sugared beverages given oral hygiene status.

higher order of fully connected nodes, equivalently not higher order interaction was detected by the estimated UGM model [e.g., 12].

## 3.3 Model verification

Statistical hypothesis tests can be carried out in order to evaluate if a simpler model with respect to the fitted UGM can be estimated, more precisely if any edge in the graph in Fig 2 can be deleted. The F-test tests whether a single variable or an interaction term can be removed from the model without information loss. The null hypothesis is that the UG model and a simpler model are statistically equivalent, so the latter is preferred because it has fewer parameters. Instead, the alternative hypothesis states that the removal of a term from the original UG model produces a significant information loss, so the UG model is preferable.

A preliminary study on the eight selected variables is also carried out within the framework of contingency table analysis and is summarized with the Fisher's exact statistical hypothesis test for count data [19]. This test is used to examine whether an association between two nominal variables in a sample is unlikely to reflect the same association in the population from which the sample is extracted.

Application of UGM on the total sample of 273 subjects and all ethnicities returns the same estimated interactions as those in the UGM in Fig 2. The results of the analysis are collected in Supplementary materials 2 and summarised in Tables 1, 2, 3, 4, S3, S4 and S5 in S2 File followed by _bis_. The analyses on the 234 subjects and on the 273 subjects lead to the same inference on the risk factors and their interactions. Thus, the conclusion we have reported in Sections 4 and 5 (Results and Discussion) seem important across different ethnic groups, but we have to acknowledge that no strong conclusions about similarities or differences between ethnicity can be drawn on the basis of this cohort.

## 4 Results

Table 2 summarises the p-values for the Fisher's exact tests for all pairs of the eight selected variables. Five marginal dependences are detected by the test and highlighted in Table 2. The

**Table 2. P-values of the Fisher's exact tests for marginal independences.**

| | Frequency of toothbrushing | Oral hygiene status | Consumption of sugared beverages | Consumption of vegetables/fruits | Breastfeeding time | Use of Pacifier | Caries incidence |
|---|---|---|---|---|---|---|---|
| Breastfeeding type | 0.586 | 0.436 | 0.374 | 0.875 | <0.001*** | 0.126 | 0.104 |
| Frequency of toothbrushing | | 0.837 | 0.613 | 0.095 | 0.453 | 0.067 | 0.787 |
| Oral hygiene status | | | <0.001*** | 0.515 | 0.057 | 0.164 | <0.001*** |
| Consumption of sugared beverages | | | | 0.718 | 0.473 | 0.361 | <0.001*** |
| Consumption of vegetables/fruits | | | | | 0.429 | 0.122 | 0.760 |
| Breastfeeding time | | | | | | 0.008** | 0.110 |
| Use of Pacifier | | | | | | | 0.824 |

P-values smaller than 0.001 suggest the four strongest relations.

* Significant (p ≤ 0.050)

** Significant (p ≤ 0.010)

*** Significant (p ≤ 0.001).

strongest ones are four: *Breastfeeding type* and *time*, *Oral hygiene status* and *Consumption of sugared beverages*, *Oral hygiene status* and *Caries incidence*, *Caries incidence* and *Consumption of sugared beverages*. The p-values for the Fisher's exact test are carried out in order to examine the most interesting three-way associations and are reported in Tables 3 and 4. Table 3 refers to independence of some pairs of study variables conditional on *Oral hygiene status*, while Table 4 is related to conditioning on *Breastfeeding time*. In contrast to the marginal associations detected in Table 2, these specific analyses on conditional subpopulations do not support any associations. Instead, they suggest independence of the three pairs of variables in the rows of Tables 3 and 4 conditional on *Oral hygiene status* and *Breastfeeding time*, respectively [20].

The F-test statistics for all sub-models of the UGM in Fig 2 are reported in S5 Table in S1 File. Only the main effect of the use of pacifier could be removed, due to a p-value greater than the reference level 0.05. But *Use of Pacifier* is statistically significant in the interaction with *Breastfeeding time*. Hence, *Use of Pacifier* is kept in the model, which is taken to be hierarchical, so as main effects are included in the model for any two-way interaction in the model. In conclusion, the estimated UGM is also supported by the F-tests and the Fisher's exact tests.

The main findings implied by the UGM in Fig 2 and the factorization in Eq (1) can be listed:

**Table 3. P-values of Fisher's exact tests for the most interesting conditional independences given the Oral hygiene status.**

| | | Oral hygiene status | |
|---|---|---|---|
| | | 1 (adequate) | 2 (not adequate) |
| Consumption of sugared beverages | Caries incidence | 0.101 | 1.000 |
| Breastfeeding time | Caries incidence | 0.505 | 0.108 |
| Breastfeeding time | Consumption of sugared beverages | 0.928 | 1.000 |

Not Significant p-values suggest independence relationships among pair of variables (left) given a specific level of Oral hygiene status. Since for all three pairs of variables the p-values are not significant for both the levels of the oral hygiene, the conclusion is that these pairs of variables are conditional independent.

**Table 4. P-values of Fisher's exact tests for the most interesting conditional independence given Breastfeeding time.**

| | | Breastfeeding time | | | |
|---|---|---|---|---|---|
| | | **0**<br>**(0 months)** | **1**<br>**(1–6 months)** | **2**<br>**(7–12 months)** | **3**<br>**(>12 months)** |
| Oral hygiene status | Use of Pacifier | 0.069 | 0.326 | 0.052 | 0.933 |
| Oral hygiene status | Breastfeeding type | 1.000 | 0.677 | 1.000 | 0.708 |
| Use of Pacifier | Breastfeeding type | 1.000 | 1.000 | 1.000 | 1.000 |

Not Significant p-values suggest independence relationships between pair of variables (left) given a specific level of Breastfeeding time. Since for all three pairs of variables the p-values are not significant for all the four levels of the time of the breastfeeding, the conclusion is that these pairs of variables are conditional independent.

1. *Consumption of vegetables/fruits* and *Frequency of toothbrushing* are marginally independent with respect to the other six variables. This implies that further analyses can be focused on the six connected variables. This factorization establishes that $X_V$, $X_B$ and $X_R$ are mutually independent random vectors, i.e. for the children we analysed, brushing teeth frequency has limited relevance for the oral hygiene status and the variation of number of carious lesions. The joint probability distribution of the six connected variables $X_R$ can be further factorized [21] according to the structure of the graph in Fig 2. This factorization is not reported here. It is sufficient for our purposes to note that each factor corresponds to an edge, nodes connecting edges play the special role of separators and that there are no cycles. Consequences of these facts are illustrated next.

2. Main conclusions can be drawn from the analysis of the six remaining variables are: *Consumption of sugared beverages* and *Caries incidence* result conditionally independent given *Oral hygiene status*, that is the effect of the consumption of sugared beverages on caries development is mediated by the oral hygiene status. Furthermore, for caries development the quality of oral hygiene is suggested to be more relevant than the frequency of toothbrushing. Conditionally on *Breastfeeding time*, the three variables *Oral hygiene status*, *Breastfeeding type* and *Use of Pacifier* result independent. In particular, *Use of Pacifier* appears to be independent from *Breastfeeding type* given *Breastfeeding time*, that is the type of breastfeeding is a risk factor for the development of caries but simply mediated by the duration of breastfeeding. In the same way, the role of the use of pacifier in the caries development is mediated by the duration of breastfeeding and it resulted statistically significant only when considered in combination with breastfeeding time.

3. Five two-way contingency tables suffice to the understanding of the relationships among the six connected variables and no higher order table is needed. This is because only five edges and no cycle are present in the UGM in Fig 2. The table matching type and time of the breastfeeding is reported on S4 Table in S1 File. It is the most troublesome for estimation because it includes five structural zeros: indeed, *Breastfeeding time* is zero for children that have not been breastfed.

4. Hard to estimate parameters are associated with cells with zero counts: the R software we used to carry out this study returns parameters estimation and the model diagnostics are satisfactory.

5. The UGM in Fig 2 has 38 parameters: 21 for two-way interactions (edges), 16 for the linear terms (nodes) and 1 for the constant term. This is a drastic reduction with respect to the $4^2 3^4 2^2$ (5184) possible combinations of the eight selected variable levels. As already stated in point 1 above, the analysis can be carried out without the two variables *Consumption of vegetables/fruit* and *Frequency of toothbrushing*.

### 4.1 Focus on caries incidence

To investigate the relevance of the seven collected variables on caries incidence, an odds-ratio analysis was carried out. The odds-ratios and the corresponding p-values are reported in S3 Table in S1 File (see also S3 Table in S2 File). It shows strong relation between caries incidence and oral hygiene status (p-value < 0.001 and odds-ratio > 50) and between caries incidence and consumption of sugared beverages (p-value < 0.001 and odds-ratio > 50). Instead, unlike what most would expect, no statistical direct evidence has been found to support a relation between the incidence of carious lesions number and toothbrushing frequency.

## 5. Discussion

The purpose of this research is to improve the understanding of the interactions among the environmental risk factors that are involved in ECC. To the best of our knowledge, this is the first caries study in which dental caries risk factors are assessed by probabilistic graphical models which are used as a tool to establish conditional independence relationships among the study variables. Log-linear models, to which UGMs relate, have been already applied in such context [15] but with a different focus. Indeed, Undirected Graphical Models (UGM) have been introduced in the statistical literature in order to model, identify and visualize conditional independence statements for random vectors [12]. A UGM can be represented by a graph with as many nodes as covariates. A missing edge between a pair of nodes means that the two variables corresponding to those two nodes are conditionally independent given all the other covariates. If a set of nodes is not connected to any other node in the graph, then the random vector of variables corresponding to that set is marginally independent from the remaining variables, and thus the joint probability distribution factorizes simplifying the analysis and the model estimation and interpretation. In its simplest form the UGM could be seen as a visualization of the more popular log-linear model. In our study UGM analysis has provided useful insights regarding specific interactions between risk factors in caries development and valuable biological information that could be missed otherwise. The analysed dataset did not contain missing values. Nevertheless, information from any incomplete observations can be incorporated into UG modelling within a full Bayesian approach or a likelihood-based approach through the EM algorithm (Expectation Maximisation) assuming the missing at random (MAR) propriety [22]. The transformation of some numerical variables in categoric ones we performed in this analysis causes a loss of information. However, this conversion is often done in order to have a more manageable dataset and to make more convenient the model. In our case, we could obtain a more direct and a more understandable interpretation of the results. We want to highlight that in such transforming process, we followed the guidelines of experts for the choices of the levels of the variables at stake. Furthermore, we also verified the robustness of our main results to different, reasonable discretizations of the numerical variables into the categorical ones.

Although sugars are recognized as having a primary role in the formation of carious lesions caries remains a multifactorial and multidimensional disease [5–7,22,23]. From a pathogenetic point of view, ECC results from complex interactions among acid-producing members of the biofilm, fermentable carbohydrates, and many host factors, including susceptible tooth surfaces and saliva. Otherwise, ECC is mainly modulated by behavioural and environmental risk factors, such as diet, fluoride exposure and oral and dental hygiene [4,23–25].

It is therefore important to know how behavioural and environmental factors are related to ECC development and what interactions between risk factors can modulate positively or negatively the ECC development in order to prepare targeted interventions for different community or group of patients. UGM represents an important opportunity to identify and efficiently

target children at high ECC risk. The careful implementation of UGM modelling can prevent misleading effect estimates and missing major and minor effects in the observational period. Moreover, UGM analysis could represent a useful tool to improve the community- and school-based oral health programs with more effective and efficient prevention strategies based on the hierarchical interactions among the ECC behavioural and environmental risk factors.

In the main part of this paper, we reported results on the dataset with 234 observations and in Supplementary Materials 2 results for the augmented sample including 39 children from twelve ethnicities. There are conflicting results in the literature regarding which ethnic groups are most susceptible to dental caries. The majority of the studies generally identify residual differences in racial/ethnic groups (in favor of one or the other ethnic groups) also after control for socioeconomic status and other confounders, both in preschool children [26,27] and in adults [28]. Even if the analysis carried out on the whole sample (273 subjects) returns substantially identical results as for the dataset with 234 subjects, a strong conclusion about similarities or differences between ethnicity can not be drawn on the basis of this cohort. Indeed the 39 excluded subjects belong to 12 different ethnic groups with great variability in dietary habits and oral hygiene. Thus, we acknowledge that our dataset is not bringing any contribution to the unresolved issues related to ethnicity and in the remainder of the paper we discuss our findings with reference to the smaller dataset.

Firstly, in our study *Consumption of sugared beverages* variable, a proxy for sugar intake, results to be conditional independent from *Caries incidence* given *Oral hygiene status* and also that *Frequency of toothbrushing* is marginally independent from the variable *Caries incidence*. This suggests that the quality of oral hygiene (and therefore the plaque index) and not the toothbrushing frequency is a primary factor that modulates (+ or -) the sugar consumption in his primary role of the ECC developer.

Secondly, in their systematic review, Kumar and Co-authors [29] reported that individuals who state that they brush their teeth infrequently are at a greater risk for the incidence or increment of new carious lesions than those brushing more frequently. But they considered only toothbrushing frequency as the main outcome because their interest was on whether toothbrushing frequency was predictive of the development of carious lesions, without any consideration for the correctness and efficiency of the oral hygiene. This introduced an important bias in the evaluation of the role of oral hygiene in caries development. In fact, intervention studies reported the success of supervised daily tooth brushing in controlling ECC development [10,30–32]. The present studies linked toothbrushing frequency, the quality of oral hygiene (checked by a senior trained dentist) and the caries development also with diet habits and made evident that the quality of oral hygiene plays a pivotal role in ECC development and that it can act as a protective factor in the ECC development. These results suggested that the community- and school-based oral health programs should focus also on the quality of oral hygiene and not only on the toothbrushing frequency. Indeed, here the odds-ratio study confirmed the findings reported in the literature: a direct relation has been found between ECC development and oral hygiene and the consumption of sugared beverages. Instead, no statistical evidence has been found to support a relation between ECC development and toothbrushing frequency. It is widely believed that effective removal of dental biofilm by toothbrushing can reduce the development of new carious lesions, but the evidence base is weak, especially because most of the results come from studies that considered the frequency of brushing and not the quality of oral hygiene [29,33].

Based on our results, the oral health education of children and parents in dentist's clinical practice and the community- and school-based oral health programs for ECC prevention should be improved with supervised toothbrushing program as reported by Macpherson et al. [10] in the results of the Scottish National supervised toothbrushing program. This is in accord

also with the Cochrane review carried out by Silva and Co-authors [9] that reported as community-based oral health promotion interventions that combine oral health education with supervised toothbrushing or professional preventive oral care can reduce dental caries in children. Other interventions, such as those that aim to promote access to fluoride, improve children's diets or provide oral health education alone, show only limited impact. Moreover, the use of fluoride supplements was reported in less than 10% of our sample because in the geographical area where our sample was selected (North West of Italy), the average of fluorine in water is estimated to be around 1 mg / l. This perhaps explains the lack of provisions and regulations regarding the artificial addition of fluorine in drinking water in Italy.

Thirdly, another group of interactions identified by the UGM is that ECC development (*Caries incidence*) is conditionally independent from *Type of breastfeeding* given *Oral hygiene status* and also from the *Breastfeeding time* and *Use of Pacifier*. In the present study the odds-ratio analysis found moderated correlation between ECC development, type of breastfeeding (p-value = 0.048 and odds-ratio = 3.329) and length of the period of breastfeeding (p-value = 0.019 and odds-ratio = 4.469, S3 Table in S1 File). Therefore, it appears that the type of breastfeeding and the use of pacifier are risk factors for the development of caries but mediated by the duration of breastfeeding and oral hygiene status, resulting both statistically significant only when considered in combination with breastfeeding time. These are interesting findings on a very debated topic in the literature [34–37]. The issue of whether bottle feeding is more cariogenic than breastfeeding is unresolved to the best of our knowledge. Some authors have not found an association between breastfeeding and dental caries, while other studies have reported the existence of such association but some authors have stated that bottle feeding is a risk factor for dental caries while another author did not find such an association [34–37]. According to a recent systematic review by Avila and Co-authors [37] breastfeeding is more effective at preventing dental caries in early childhood than bottle feeding, although the duration of breast- or bottle-feeding in the studies analysed could not be determined. So, among the ECC risk factors not only the type, but also the duration of feeding represents a critical issue in literature. The UGM again displayed on this topic the pivotal role of the quality of oral hygiene (this information is lost in commonly used statistical models) and how it is able to act as a protective factor in ECC development (Fig 2). Thus, in order to decrease the ECC incidence, the antenatal and postnatal educational interventions to mothers for breastfeeding practices (focusing on the duration of the breastfeeding period, which has been shown to be one of the most important risk factors), need to be supported by incorporating mother and child oral health promotion to reduce caries experience, improve oral hygiene and dietary habits, as is supported by the results of the systematic review by Abou El Fadl and Co-authors [36].

The major strength of this study is the implementation of UG modelling into a dental/caries dataset. Although this study did not identify new biological relationships, its unique contribution is in the interpretation of the findings because it allows the identification of a possible hierarchy that regulates the interaction between risk factors for caries development and because it underlined the pivotal role of the correctness of oral hygiene. We have also identified the most significant interactions in ECC development to improve the efficiency and efficacy of the community- and school-based oral health programs for ECC prevention. The key findings are effect of sugar intake (*Consumption of sugared beverages*) on caries incidence is mediated by the oral hygiene; for the caries incidence, oral hygiene quality is more relevant than toothbrushing time. The use of pacifier and the type of breastfeeding are not directly involved in ECC development, but they become significant risk factors when associated with incorrect breastfeeding practices. Above all, prolonged breastfeeding is one of the main risk factors for dental caries development.

## Supporting information

**S1 File.**
(DOCX)

**S2 File.**
(DOCX)

## Author Contributions

**Conceptualization:** Alessandro Ugolini, Paola Agostino, Armando Silvestrini-Biavati.

**Data curation:** Francesco Porro, Federico Carli.

**Formal analysis:** Francesco Porro, Federico Carli, Eva Riccomagno.

**Investigation:** Alessandro Ugolini, Paola Agostino.

**Software:** Francesco Porro, Federico Carli, Eva Riccomagno.

**Supervision:** Armando Silvestrini-Biavati, Eva Riccomagno.

**Visualization:** Federico Carli.

**Writing – original draft:** Alessandro Ugolini.

**Writing – review & editing:** Armando Silvestrini-Biavati, Eva Riccomagno.

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
