## [Decision Letter · Decision Letter 0]

5 Sep 2022

PONE-D-22-04095Probabilistic graphical modelling of early childhood caries developmentPLOS ONE

Dear Dr. Ugolini,

Thank you for submitting your manuscript to PLOS ONE. After careful consideration, we feel that it has merit but does not fully meet PLOS ONE’s publication criteria as it currently stands. Therefore, we invite you to submit a revised version of the manuscript that addresses the points raised during the review process.

Your manuscript has been assessed by two peer-reviewers and their reports are appended below. 

The reviewers comment that the choices of statistical approach require further clarification. In addition, the reviewers have raised major concern with the exclusion of almost 14% of the sample due to missing data, suggesting that this exclusion may lead to bias, which has not been appropriately addressed or discussed in the study. 

Could you please revise the manuscript to carefully address the concerns raised?

We look forward to receiving your revised manuscript.

Kind regards,

Maria Elisabeth Johanna Zalm, Ph.D

Editorial Office

PLOS ONE

Journal Requirements:

a) Did participants provide their written or verbal informed consent to participate in this study?

Reviewers' comments:

Reviewer's Responses to Questions

**Comments to the Author**

1. Is the manuscript technically sound, and do the data support the conclusions?

Reviewer #1: Partly

Reviewer #2: Yes

2. Has the statistical analysis been performed appropriately and rigorously? 

Reviewer #1: No

Reviewer #2: Yes

3. Have the authors made all data underlying the findings in their manuscript fully available?

Reviewer #1: No

Reviewer #2: Yes

4. Is the manuscript presented in an intelligible fashion and written in standard English?

Reviewer #1: No

Reviewer #2: Yes

5. Review Comments to the Author

Reviewer #1: This generally well-written paper presents an undirected graphical model (UGM) to analyze early childhood caries (ECC) data of 234 children longitudinally from when they were 3 to 5 years old. However, the UGM is fit with a log-linear model of 8 variables including caries where the measures have been categorized over time and from their original scales to a smaller number of categories, mostly with little justification. The authors claim this as the first UGM applied to longitudinal pediatric dentistry, but the UGM they present is a log-linear model which have been applied to the topic since at least 1983 (Oler & Bentley, Stat Med). The UGM they present does help elucidate an interesting and possibly novel correlation pattern among the variables, identifying how they are intercorrelated. However, there are some important details that need to be added or better clarified.

Comments are numbered for convenience; they are not in order of importance.

1. ECC CAN begin earlier than 3 years of age (eg 1 year old). AAPD has promoted the phrase "2 is too late" to indicate children’s first dental visit at 2nd birthday is too late because ECC starts before 2 years of age, at least in certain communities.

so

either change to

"ECC can begin early in life, as early as 1 year old,"

or

"ECC typically begins early...around 3 years old"

2. “Caries” like “diabetes” is the name of the disease and is a singular verb not plural despite ending in an “s”. there is no “number of caries” just as there is no “number of diabetes” but rather “number of carious lesions” or “number of cavities” (just as there might be number of diabetes wounds).

3. What are “cliques”? is this the same as “subsets”? if so, use the term “subsets”. if not, this is a non-standard term, so please define (such as “intercorrelated variables”).

4. What are “problem” variables? this is just the same as “study” variables, right? If so, delete “problem” throughout as the term is more confusing than clarifying. if not, this is a non-standard term, so please define.

5. How do UGM / MRF models improve upon other types of models? Why are other models such as latent class logistic models or structural equation models (SEM) insufficient, particularly because the UGM in this paper is a log-linear model which has been used in multiple longitudinal pediatric dentistry studies (eg Oler & Bentley, 1983)?

6. Using and reporting the caries calibration procedures and results are helpful; among the 15 children, how many caries lesions or fillings were observed at the calibration? If 99% of tooth surfaces are sound (not caries lesions or fillings), this can inflate even chance-adjusted kappas.

7. Unfortunately, this analysis reduces the number of caries lesions/fillings at 3 timepoints (years) into a simple dichotomy of developed any new carious lesions/fillings vs did not develop any. The analysis does not distinguish between those with 0, 1, or 2+ new carious lesions/lesions or between those who develop 1+ by age 4 vs those between 4 and 5. “Caries variation” would be much clearer as “caries incidence”.

8. 45 (18+12+11+4) or almost 14% of 325 children were excluded because of missing data (incomplete parent questionnaires, having nonstandard number or shape of teeth, or only having 1 dental exam, plus 1 drop out and 3 missing “important” factors. This can lead to bias. Did these children differ from others in terms of ECC or explanatory variables?

9. The issue of handling missing data is not mentioned except to say UGMs can accommodate missing data but then in contrast listwise deletion is used on nearly 14% of the children. Missing data do not appear to be properly handled by either using a method that makes modest missing data assumptions or using a method like multiple imputation to include data of children with incomplete data. What missing data assumptions to UGMs and these log-linear models use? What are the implications?

10. As written, it isn’t initially clear that N=234 results from dropping the 39 children from 12 ethnicities. Implicit in the comments about ethnicity it seems is that they are non-Italian or non-Western European ethnicity. It would be clearer to specify this more explicitly – who does the non-ethnic group of 234 include? Since there is not much difference between the results with N=273 and with N=234, why not use N=273 as primary? Do they have somewhat greater precision (or power)?

11. In 3.1, the variable level labeling is inconsistent, sometimes beginning with 0 and sometimes 1.

12. In 3.1, Is toothbrushing measuring how often the children brush their own teeth each day or how often adults (parents) brush the children’s teeth daily? 3-5 year olds are not capable of properly brushing their own teeth. If this is really number of times children brush their own teeth it is not surprising this variable is unimportant.

13. In 3.1, the distinctions between daily, weekly, and occasionally are not clear since they do not use mutually exclusive and exhaustive ranges like other variables such as breastfeeding time. Perhaps measuring vegetable and fruit consumption in such as crude way results in that variable seeming unrelated to any others. In contrast, sugar-sweetened beverages have such a large and dramatic effect that even in this crude way the effects are evident.

14. In 3.1, how is oral hygiene adequacy defined/operationalized? It is not until the Discussion section that it is mentioned that a senior dental clinician checked oral hygiene quality.

15. In 3.2, state that the combination of possible combinations of values is the found by multiplying the number of levels of each of the 8 variables in 3.1. Add that this results in 5184 possible combinations.

16. How does the UGM model differ from a mediation model (eg SEM) where sugar-sweetened beverages, breastfeeding (type and time) and pacifier use relate to oral hygiene status (the mediator) which relates to caries incidence?

17. Figure 2 does not show any arcs, just straight lines (edges); is this what is meant by “arc”? it would be clearer if the lines connecting nodes were curved or if the term “arc” was not used but “edge” was used instead.

18. Although the authors appear to have made the data available in Mendeley, the dataset could not be found so availability could not be confirmed.

19. This paper appears to rely too much on p-values.

The American Statistician March, 2019 special issue entitled “Statistical Inference in the 21st Century: A World Beyond p<0.05” (https://www.tandfonline.com/toc/utas20/73/sup1) contained 43 papers presenting a multitude of widely varying viewpoints and recommendations for modern statistical inference. Among the few consensus items across all the papers were:

• “Don’t base your conclusions solely on whether an association or effect was found to be ‘statistically significant’ (i.e., the p-value passed some arbitrary threshold such as p < 0.05).

• Don’t believe that an association or effect exists just because it was statistically significant.

• Don’t believe that an association or effect is absent just because it was not statistically significant.

• Don’t believe that your p-value gives the probability that chance alone produced the observed association or effect or the probability that your test hypothesis is true.

• Don’t conclude anything about scientific or practical importance based [solely] on statistical significance (or lack thereof).”

The overall summary from the special issue is: “Accept uncertainty. Be thoughtful, open, and modest [ATOM].” (Wasserstein, Schirm, Lazar 2019). The authors are encouraged to review these issues and modify the interpretations and language in this paper as appropriate.

In addition, there are some minor editorial items that require correction:

1. “a-prior” should be “a-priori”

2. “…values, they supply a compact” should be “…values, supply a compact” to maintain the parallelism of the earlier sentence (“are, can, supply…”)

3. T0, T1, T2 are defined as abbreviations but never used again; either using the abbreviations again or delete them

4. Be consistent with use of X and X [italicized]

5. “One Down child” should be “One child with Down’s syndrome…”

6. 3.1 3. States that caries was measured by 4 original variables but aren’t there only 3; i.e. Number of lesions/restorations at age 3, age 4, and age 5?

7. In Table 1, add counts in parentheses in each cell; round percentages to 1 decimal place.

8. “Otherwise, ECC are mainly modulated…” should be “Otherwise, ECC is mainly modulated…”

REF

Oler J, Bentley JM. Log-linear model selections in a rural dental health study. Stat Med. 1983 Jan-Mar;2(1):59-69. doi: 10.1002/sim.4780020107. PMID: 6648120.

Reviewer #2: The article is well-written. The statistical analysis of the paper, which is mainly based on undirected graphical models, is well done and described. However, as a major comment, some of the considered variables are, in principle, of numerical type. Examples are: “Use of pacifier”, “Breastfeeding time”, and “Frequency of toothbrushing”. So, transforming them as nominal variables amounts to a twofold loss of information: 1) from numerical to ordinal, and 2) from ordinal to nominal. If these variables had been left in their original nature, the authors could have found an appropriate statistical model that would allow them to take advantage of this additional information. Authors need to discuss this aspect to convince the reader about their choice.

Below are some typos to be corrected.

-- Section 3.1: replace “273 and 234 observations identifies the same risk” with “273 and 234 observations identify the same risk”

-- Section 3.1: replace “(see also the section Model Verification)” with “(see also Section 3.3)”

-- Conclusions: replace “with a more effective and efficient prevention strategies” with “with more effective and efficient prevention strategies”

-- Figure 1: replace “3<years>4” with “3<years<4”.< p=""></years<4”.<></years>

6. PLOS authors have the option to publish the peer review history of their article (what does this mean?). If published, this will include your full peer review and any attached files.

Reviewer #1: No

Reviewer #2: No

---

## [Author Response · Author response to Decision Letter 0]

22 Dec 2022

Answers to the Referee Report 1 on the paper:

Probabilistic graphical modelling of early childhood caries development (PONE-D-22-04095)

First of all, we would like to thank the Referee for his/her careful review of the paper. We did our best to revise the article according to his/her valuable suggestions, for which we are grateful. Please find below our answers (in blue) to the comments raised by the Referee. All the changes in the manuscript are in red. Please note that we added an author who recently started to collaborate with us and whose collaboration was instrumental to the production of this revision.

This generally well-written paper presents an undirected graphical model (UGM) to analyze early childhood caries (ECC) data of 234 children longitudinally from when they were 3 to 5 years old. However, the UGM is fit with a log-linear model of 8 variables including caries where the measures have been categorized over time and from their original scales to a smaller number of categories, mostly with little justification. The authors claim this as the first UGM applied to longitudinal pediatric dentistry, but the UGM they present is a log-linear model which have been applied to the topic since at least 1983 (Oler & Bentley, Stat Med). The UGM they present does help elucidate an interesting and possibly novel correlation pattern among the variables, identifying how they are intercorrelated. However, there are some important details that need to be added or better clarified.

Thanks for indicating the paper by Oler & Bentley (1983). It is indeed a very interesting reading and pertinent to our manuscript. We were not aware of it and, actually, had some difficulties in retrieving it. We reviewed our manuscript to acknowledge this earlier work and use of the log-linear models in longitudinal pediatric dentistry. Notice that UGM provides a graphical representation which helps in visualizing and communicating independences and mediating variables and thus support clinical interpretation of the results. Oler & Bentley (1983) dealt with a specific clinical question different from the one in our manuscript. 

ECC CAN begin earlier than 3 years of age (eg 1 year old). AAPD has promoted the phrase "2 is too late" to indicate children’s first dental visit at 2nd birthday is too late because ECC starts before 2 years of age, at least in certain communities.

So either change to

"ECC can begin early in life, as early as 1 year old,"

or

"ECC typically begins early...around 3 years old"

The sentence has been modified as proposed. 

“Caries” like “diabetes” is the name of the disease and is a singular verb not plural despite ending in an “s”. there is no “number of caries” just as there is no “number of diabetes” but rather “number of carious lesions” or “number of cavities” (just as there might be number of diabetes wounds).

As remarked, “number of caries” has been replaced by “number of carious lesions”.

What are “cliques”? is this the same as “subsets”? if so, use the term “subsets”. if not, this is a non-standard term, so please define (such as “intercorrelated variables”).

The cliques are subsets of vertices such that every two distinct vertices are adjacent. As proposed, the definition has been added in the manuscript.

What are “problem” variables? this is just the same as “study” variables, right? If so, delete “problem” throughout as the term is more confusing than clarifying. if not, this is a non-standard term, so please define.

As remarked, the labels “problem variable” have been replaced by “study variable”.

How do UGM / MRF models improve upon other types of models? Why are other models such as latent class logistic models or structural equation models (SEM) insufficient, particularly because the UGM in this paper is a log-linear model which has been used in multiple longitudinal pediatric dentistry studies (eg Oler & Bentley, 1983)?

The UGM are models that allow the understanding of the conditional independence among observed variables. We believe that an important advantage that such models share with others (for example with the aforementioned SEM) is the possibility to easily interpret the results and to graphically represent the relationships among the variables. Furthermore, for using UGM we do not need to identify a specific measured variable as a response variable. We decided to use UGM because the aim of our paper is not the investigation on the changes of a (dependent) response variable depending on some covariates, but it is the identification (and the corresponding interpretation) of the interactions among all the considered variables. 

You are absolutely right, also in Oleg & Bentley 1983 a log-linear model has been used to analyze a pediatric dentistry study, but in that case, the study variables and the clinical question were very different from those we used in our paper (cfr for example the Grade of School, the Dental Delivery Mode, and the Oral Health Care Utilization Pattern, used in Oleg & Bentley 1983). 

Finally, as far as we know, the number of applications related to UGM in biological/medical fields is fast increasing in the last years and we believe that the dissemination of different methodologies over different scientific areas is very worthwhile. 

Using and reporting the caries calibration procedures and results are helpful; among the 15 children, how many caries lesions or fillings were observed at the calibration? If 99% of tooth surfaces are sound (not caries lesions or fillings), this can inflate even chance-adjusted kappas.

As suggested, we checked and we found that at each examiner calibration, the percentage of children with caries lesions or filling was much greater than 1%. In Section 2.3 we described the double calibration process we adopted: first at the WHO Centre of Collaboration for Epidemiology and Community Dentistry (University of Milan) and then a second calibration as indicated in the study. 

Reference: Castiglia P, Campus G, Solinas G, Maida C, Strohmenger L. Children's oral health in Italy: training and clinical calibration of examiners for the National Pathfinder about caries disease. Oral Health Prev Dent. 2007;5(4):255-61. PMID: 18173085.

Unfortunately, this analysis reduces the number of caries lesions/fillings at 3 timepoints (years) into a simple dichotomy of developed any new carious lesions/fillings vs did not develop any. The analysis does not distinguish between those with 0, 1, or 2+ new carious lesions/lesions or between those who develop 1+ by age 4 vs those between 4 and 5. “Caries variation” would be much clearer as “caries incidence”.

In this analysis we decided to distinguish the children into only two groups: those that developed any new carious lesions/fillings and those that did not develop any. Number of caries lesions depends on variables which are not part of this study such as teeth enamel, but this is a controversial issue in the literature and won’t be dealt with here. As suggested, “Caries variation” has been replaced by “Caries incidence”.

45 (18+12+11+4) or almost 14% of 325 children were excluded because of missing data (incomplete parent questionnaires), having nonstandard number or shape of teeth, or only having 1 dental exam, plus 1 drop out and 3 missing “important” factors. This can lead to bias. Did these children differ from others in terms of ECC or explanatory variables?

For the 38 subjects, which have at least a visit, there are not significant differences in terms of ECC and explanatory variables. For the other excluded subjects, these differences cannot be ascertained because of the missing information.

The issue of handling missing data is not mentioned except to say UGMs can accommodate missing data but then in contrast listwise deletion is used on nearly 14% of the children. Missing data do not appear to be properly handled by either using a method that makes modest missing data assumptions or using a method like multiple imputation to include data of children with incomplete data. What missing data assumptions to UGMs and these log-linear models use? What are the implications?

As argued in the previous point, we considered a clean dataset without missing data. The reason is that we preferred to not introduce more uncertainty in the dataset by using for example multiple imputation methods) to replace missing data.

As written, it isn’t initially clear that N=234 results from dropping the 39 children from 12 ethnicities. Implicit in the comments about ethnicity it seems is that they are non-Italian or non-Western European ethnicity. It would be clearer to specify this more explicitly – who does the non-ethnic group of 234 include? Since there is not much difference between the results with N=273 and with N=234, why not use N=273 as primary? Do they have somewhat greater precision (or power)?

As detailed in Section 5, in this paper we decided to discuss our findings with reference to the smallest dataset with N=234 children because we acknowledge that our datasets are not adequate for drawing strong conclusions on similarities or differences between different ethnicities. As described in the manuscript, we performed the analysis also for the N=273 children and we did not find much difference between the two results. Nevertheless, we are aware that our analysis cannot contribute to the unresolved issue in the literature about which ethnic groups are more susceptible to dental carious lesions. For these reasons we excluded 39 subjects belonging to 12 different ethnic groups with great variability in dietary habits and oral hygiene.

In 3.1, the variable level labeling is inconsistent, sometimes beginning with 0 and sometimes 1.

The values assumed by the variables are displayed in Table 1_bis (in the Supplementary material 2). As such table shows, the value 0 has been used for some variables only to represent the absence of the corresponding “phenomenon”: for example, Breastfeeding Type is 0 only if the child was not breastfed, or Use of Pacifier is 0 only if the child did not use it. 

The variables beginning with 1 (for example Frequency of Toothbrushing, Consumption of Sugared Beverages, and Consumption of Vegetables/fruits), are those where there is not a level corresponding to the absence of the “phenomenon” (all the children toothbrush at least once a day, all the children drink sugared beverages at least occasionally, all the children eat a portion of vegetables or fruits at least occasionally, etc...). 

In 3.1, the distinctions between daily, weekly, and occasionally are not clear since they do not use mutually exclusive and exhaustive ranges like other variables such as breastfeeding time. Perhaps measuring vegetable and fruit consumption in such as crude way results in that variable seeming unrelated to any others. In contrast, sugar-sweetened beverages have such a large and dramatic effect that even in this crude way the effects are evident.

We understand the point raised by the referee. Nevertheless, the full questions asked in the questionnaire (in Italian) did not give rise to ambiguous interpretation.

In 3.1, how is oral hygiene adequacy defined/operationalized? It is not until the Discussion section that it is mentioned that a senior dental clinician checked oral hygiene quality.

In Section 2.3 a detailed description of the procedure for evaluating the Oral Hygiene Adequacy is provided. For clarity, we added in Section 3.1 a sentence mentioning it.

In 3.2, state that the combination of possible combinations of values is the found by multiplying the number of levels of each of the 8 variables in 3.1. Add that this results in 5184 possible combinations.

As requested, the result of 5184 has been added also in section 3.1 of the manuscript.

How does the UGM model differ from a mediation model (eg SEM) where sugar-sweetened beverages, breastfeeding (type and time) and pacifier use relate to oral hygiene status (the mediator) which relates to caries incidence?

A mediation model is an analysis that seeks to identify the mechanism that underlies an observed relationship between an independent variable and a dependent variable, via the inclusion of a third explanatory variable, known as a mediator variable. In our analysis we do not consider a dependent (response) variable, since the aim of our analysis is the investigation on the interactions among all the observed variables.

Figure 2 does not show any arcs, just straight lines (edges); is this what is meant by “arc”? it would be clearer if the lines connecting nodes were curved or if the term “arc” was not used but “edge” was used instead.

As suggested, in the manuscript the term “arc” has been replaced by “edge”.

Although the authors appear to have made the data available in Mendeley, the dataset could not be found so availability could not be confirmed.

Clearly something went wrong in the original submission of the dataset. We uploaded it again in Mendeley.

“Early childhood caries development”, Mendeley Data, V1, doi: 10.17632/7rsxyfsnd5.1 

Direct link: https://data.mendeley.com/datasets/7rsxyfsnd5

This paper appears to rely too much on p-values. The American Statistician March, 2019 special issue entitled “Statistical Inference in the 21st Century: A World Beyond p<0.05” (https://www.tandfonline.com/toc/utas20/73/sup1) contained 43 papers presenting a multitude of widely varying viewpoints and recommendations for modern statistical inference. Among the few consensus items across all the papers were:

- “Don't base your conclusions solely on whether an association or effect was found to be ‘statistically significant’ (i.e., the p-value passed some arbitrary threshold such as p<0.05).

- Don’t believe that an association or effect exists just because it was statistically significant.

- Don’t believe that an association or effect is absent just because it was not statistically significant.

- Don’t believe that your p-value gives the probability that chance alone produced the observed association or effect or the probability that your test hypothesis is true.

- Don’t conclude anything about scientific or practical importance based [solely] on statistical significance (or lack thereof).”

The overall summary from the special issue is: “Accept uncertainty. Be thoughtful, open, and modest [ATOM].” (Wasserstein, Schirm, Lazar 2019). The authors are encouraged to review these issues and modify the interpretations and language in this paper as appropriate.

As suggested, the language and the interpretations have been modified in some points, accordingly to the aforementioned issues. Clearly, we tested our conclusions in different ways for example fitting other models and testing the robustness to a slight variation of the dataset (leave-one-out). Again, we chose to present in a paper the UGM model because it summarizes our findings in a holistic, clear picture.

In addition, there are some minor editorial items that require correction:

1. “a-prior” should be “a-priori”

As suggested, the word has been corrected.

2. “…values, they supply a compact” should be “…values, supply a compact” to maintain the parallelism of the earlier sentence (“are, can, supply…”)

The sentence has been modified, as suggested.

3. T0, T1, T2 are defined as abbreviations but never used again; either using the abbreviations again or delete them

As suggested, the abbreviations have been removed.

4. Be consistent with use of X and X [italicized]

The notation of X is now consistent.

5. “One Down child” should be “One child with Down’s syndrome…”

The sentence has been modified, as suggested.

6. 3.1 3. States that caries was measured by 4 original variables but aren’t there only 3; i.e. Number of lesions/restorations at age 3, age 4, and age 5?

The sentence has been modified, as suggested.

7. In Table 1, add counts in parentheses in each cell; round percentages to 1 decimal place.

As proposed, in Table 1 the counts have been added and the percentages rounded.

8. “Otherwise, ECC are mainly modulated…” should be “Otherwise, ECC is mainly modulated…”

The sentence has been modified, according to the suggestion.

REF

Oler J, Bentley JM. Log-linear model selections in a rural dental health study. Stat Med. 1983 Jan-Mar;2(1):59-69. doi: 10.1002/sim.4780020107. PMID: 6648120.

Answers to the Referee Report 2 on the paper:

Probabilistic graphical modelling of early childhood caries development (PONE-D-22-04095)

First of all, we would like to thank the Referee for his/her careful review of the paper. We did our best to revise the article according to his/her valuable suggestions, for which we are grateful. Please find below our answers (in blue) to the comments raised by the Referee. All the changes in the manuscript are in red. Please note that we added an author who recently started to collaborate with us and whose collaboration was instrumental to the production of this revision.

The article is well-written. The statistical analysis of the paper, which is mainly based on undirected graphical models, is well done and described. However, as a major comment, some of the considered variables are, in principle, of numerical type. Examples are: “Use of pacifier”, “Breastfeeding time”, and “Frequency of toothbrushing”. So, transforming them as nominal variables amounts to a twofold loss of information: 1) from numerical to ordinal, and 2) from ordinal to nominal. If these variables had been left in their original nature, the authors could have found an appropriate statistical model that would allow them to take advantage of this additional information. Authors need to discuss this aspect to convince the reader about their choice.

You are absolutely right: transforming numerical variables in categoric ones causes a loss of information. However, this conversion is often done in order in order to have a more manageable dataset and to make more convenient the model. In our case, we could obtain a more direct and a more understandable interpretation of the results. We want to highlight that in such transforming process, we followed the guidelines of experts for the choices of the levels of the variables at stake. Furthermore, we also verified the robustness of our main results to different, reasonable discretizations of the numerical variables into the categorical ones. 

Section 3.1: replace “273 and 234 observations identifies the same risk” with “273 and 234 observations identify the same risk”

The sentence has been modified as suggested.

Section 3.1: replace “(see also the section Model Verification)” with “(see also Section 3.3)”

The sentence has been modified as suggested.

Conclusions: replace “with a more effective and efficient prevention strategies” with “with more effective and efficient prevention strategies”

The sentence has been corrected.

Figure 1: replace “34” with “3<4”.< p=""><4”.<>

 The errors in Figure 1 have been corrected.

---

## [Decision Letter · Decision Letter 1]

14 Jun 2023

PONE-D-22-04095R1Probabilistic graphical modelling of early childhood caries developmentPLOS ONE

Dear Dr. Ugolini,

Thank you for submitting your manuscript to PLOS ONE. After careful consideration, we feel that it has merit but does not fully meet PLOS ONE’s publication criteria as it currently stands. Therefore, we invite you to submit a revised version of the manuscript that addresses the points raised during the review process.

We look forward to receiving your revised manuscript.

Kind regards,

Raheel Allana

Academic Editor

PLOS ONE

Reviewers' comments:

Reviewer's Responses to Questions

**Comments to the Author**

1. If the authors have adequately addressed your comments raised in a previous round of review and you feel that this manuscript is now acceptable for publication, you may indicate that here to bypass the “Comments to the Author” section, enter your conflict of interest statement in the “Confidential to Editor” section, and submit your "Accept" recommendation.

Reviewer #1: (No Response)

Reviewer #2: All comments have been addressed

2. Is the manuscript technically sound, and do the data support the conclusions?

Reviewer #1: Yes

Reviewer #2: Yes

3. Has the statistical analysis been performed appropriately and rigorously? 

Reviewer #1: Yes

Reviewer #2: Yes

4. Have the authors made all data underlying the findings in their manuscript fully available?

Reviewer #1: Yes

Reviewer #2: Yes

5. Is the manuscript presented in an intelligible fashion and written in standard English?

Reviewer #1: Yes

Reviewer #2: Yes

6. Review Comments to the Author

Reviewer #1: The revised manuscript largely addressed the issues the reviewers identified. However, some items remain inadequately addressed.

1. In the response to Reviewer #1, the authors say they are using listwise deletion of missing data for simplicity but in the paper they mention that UGMs can accommodate missing data but do not mention in the discussion how this could be accomplished in UGMs. What missing data assumptions (eg assuming data are Missing Completely at Random or data are Missing at Random) do UGMs make? In addition, the paper would be much more useful if the authors – perhaps in the Discussion section – explain how this UGM can be expanded to accommodate missing data.

2. In the response to Reviewer #1, the authors note their preference to report results with N=234 after dropping the 39 children from 12 ethnicities. Although the authors have clearly stated who is excluded in the n=39, they have not clearly defined who is included in the N=234. Clearly state who the non-”ethnic” group of 234 includes; is it Italian-born children, children of Italian-born parents, children of Southern European parents, something else?

3. The revised paper made no changes to rely less on p-values as Reviewer #1 requested – Tables 2,3,4 only present p-values with no measures of strength of association such as odds ratios (and 95% confidence intervals) or correlations which would convey the information much better to readers. Adding some measure of strength of association (ORs or correlations) as labels on the edges of the graph in Figure 2 would be very helpful with understanding and interpreting the results.

4. Strongly suggest removing p-values in the text corresponding to kappas which only test that kappas differ from 0; even kappas below researchers’ thresholds for adequate agreement almost always significantly exceed 0, so the p-values are not useful.

5. In the response to Reviewer #1, the authors wrote “As proposed, in Table 1 the counts have been added and the percentages rounded.” But Table 1 has not been modified in the revision.

6. Reviewer #2 identified a “major comment” about impacts of reducing scale from numeric to ordinal to dichotomous. The response explains this was done for convenience to obtain a manageable dataset. However, they do not address the reviewers’ observation that information loss occurs. This should at least be added to the Discussion.

Some typographical errors require attention:

a. “A kappa values for…” should be “Kappa values for…”

b. Further edit the paragraph to refer to incidence instead of variation. Change text from

“…zero if there is no variation…”

to

“…zero if there is no increment or negative increment...”

c. In Figure 2, change from “CARIES VARIATION” to “CARIES INCIDENCE”

Reviewer #2: (No Response)

7. PLOS authors have the option to publish the peer review history of their article (what does this mean?). If published, this will include your full peer review and any attached files.

Reviewer #1: No

Reviewer #2: No

---

## [Author Response · Author response to Decision Letter 1]

19 Jul 2023

PONE-D-22-04095R1

Probabilistic graphical modelling of early childhood caries development

PLOS ONE

1. In the response to Reviewer #1, the authors say they are using listwise deletion of missing data for simplicity but in the paper they mention that UGMs can accommodate missing data but do not mention in the discussion how this could be accomplished in UGMs. What missing data assumptions (eg assuming data are Missing Completely at Random or data are Missing at Random) do UGMs make? In addition, the paper would be much more useful if the authors – perhaps in the Discussion section – explain how this UGM can be expanded to accommodate missing data.

Thanks for the comment. The MAR property is a key condition for the validity of inference based on the likelihood. When it does not hold, estimates will be biased. The MAR assumption is difficult to verify in practice and its plausibility relies on substantive reasonableness. However, as demonstrated in [Edwards, 2000], "It is possible to evaluate the plausibility of the MAR property, even without detailed knowledge of the missing data process.”

As suggested, in the Discussion section, a sentence about the issue of missing data has been added in the manuscript. 

2. In the response to Reviewer #1, the authors note their preference to report results with N=234 after dropping the 39 children from 12 ethnicities. Although the authors have clearly stated who is excluded in the n=39, they have not clearly defined who is included in the N=234. Clearly state who the non-”ethnic” group of 234 includes; is it Italian-born children, children of Italian-born parents, children of Southern European parents, something else?

The considered N=234 children are Italian-born ones, with Italian-born parents. As suggested, a sentence about this point has been added in the Data preparation section. 

3. The revised paper made no changes to rely less on p-values as Reviewer #1 requested – Tables 2,3,4 only present p-values with no measures of strength of association such as odds ratios (and 95% confidence intervals) or correlations which would convey the information much better to readers. Adding some measure of strength of association (ORs or correlations) as labels on the edges of the graph in Figure 2 would be very helpful with understanding and interpreting the results.

Thank you for your comment. We performed the Chi-squared independence test for all the pairs of variables that our model assesses as dependent (that is, linked by an edge), by considering the corresponding contingency tables. 

Here is the procedure for the variables “Caries Incidence” and “Oral Hygiene Status”: 

The contingency table is:

 Oral Hygiene Status

Caries Incidence 1 2

0 187 1

1 10 36

And the value of the Test Statistic Chi-squared =161.95 (with df=1), 

and p-value<2.2 e-16

The results for all the other pairs are included in the following table, and they confirm the results of our model: all these variable pairs can be considered as dependent.

Variable1 Variable2 Test Stat. Chi-squared value df p-value

Oral Hygiene Status Consump. of Sugar Beverages 190.45 2 <2.2 e-16

Oral Hygiene Status Breastfeeding Time 8.3226 3 0.03979

Breastfeeding Type Breastfeeding Time

 236.48 6 <2.2 e-16

Use of pacifier Breastfeeding Time

 22.686 9 0.006941

Nevertheless, we decided not to include these results in the paper since they can be misunderstood since they are related to tests of independence, while our model is a conditional dependency one. Moreover, in Probabilistic Graphical Models, it is used to compare different models by using the likelihood function, but this is not the case in our study. For these reasons, we finally decided to leave the Tables with the p-values, as we believe that the interest readers can be familiar with them. 

4. Strongly suggest removing p-values in the text corresponding to kappas which only test that kappas differ from 0; even kappas below researchers’ thresholds for adequate agreement almost always significantly exceed 0, so the p-values are not useful.

As suggested, the p-values related to kappas have been removed from the manuscript.

5. In the response to Reviewer #1, the authors wrote “As proposed, in Table 1 the counts have been added and the percentages rounded.” But Table 1 has not been modified in the revision.

We are sorry, but in the submission process of the revised manuscript, we lost the new version of Table 1. Now the correct version of Table 1 has been added.

6. Reviewer #2 identified a “major comment” about impacts of reducing scale from numeric to ordinal to dichotomous. The response explains this was done for convenience to obtain a manageable dataset. However, they do not address the reviewers’ observation that information loss occurs. This should at least be added to the Discussion.

As suggested, a sentence about the information loss due to the variable trasformation (from ordinal to dichotomous) has been added in the Discussion section.

7. Some typographical errors require attention:

a. “A kappa values for…” should be “Kappa values for…”

b. Further edit the paragraph to refer to incidence instead of variation. Change text from

“…zero if there is no variation…”

to

“…zero if there is no increment or negative increment...”

c. In Figure 2, change from “CARIES VARIATION” to “CARIES INCIDENCE”

As suggested, we revised all the manuscript. 

We fixed the aforementioned typographical errors and other ones we found in the revision process. 

As suggested, the label in Figure 2 has been changed.

---

## [Decision Letter · Decision Letter 2]

10 Oct 2023

Probabilistic graphical modelling of early childhood caries development

PONE-D-22-04095R2

Dear Dr. Ugolini,

We’re pleased to inform you that your manuscript has been judged scientifically suitable for publication and will be formally accepted for publication once it meets all outstanding technical requirements.

Kind regards,

Francisco Wanderley Garcia de Paula-Silva, DDS, MSc, PhD

Academic Editor

PLOS ONE

Additional Editor Comments (optional):

The authors addressed properly the questions raised by the reviewers.

Reviewers' comments:

Reviewer's Responses to Questions

**Comments to the Author**

1. If the authors have adequately addressed your comments raised in a previous round of review and you feel that this manuscript is now acceptable for publication, you may indicate that here to bypass the “Comments to the Author” section, enter your conflict of interest statement in the “Confidential to Editor” section, and submit your "Accept" recommendation.

Reviewer #1: All comments have been addressed

Reviewer #2: All comments have been addressed

2. Is the manuscript technically sound, and do the data support the conclusions?

Reviewer #1: Yes

Reviewer #2: Yes

3. Has the statistical analysis been performed appropriately and rigorously? 

Reviewer #1: Yes

Reviewer #2: Yes

4. Have the authors made all data underlying the findings in their manuscript fully available?

Reviewer #1: Yes

Reviewer #2: Yes

5. Is the manuscript presented in an intelligible fashion and written in standard English?

Reviewer #1: Yes

Reviewer #2: Yes

6. Review Comments to the Author

Reviewer #1: the authors have adequately addressed all the comments.

Reviewer #2: (No Response)

7. PLOS authors have the option to publish the peer review history of their article (what does this mean?). If published, this will include your full peer review and any attached files.

Reviewer #1: No

Reviewer #2: No

---

## [Editor Report · Acceptance letter]

20 Oct 2023

PONE-D-22-04095R2 

Probabilistic graphical modelling of early childhood caries development 

Dear Dr. Ugolini:

I'm pleased to inform you that your manuscript has been deemed suitable for publication in PLOS ONE. Congratulations! Your manuscript is now with our production department. 

Kind regards, 

on behalf of

Prof. Dr. Francisco Wanderley Garcia de Paula-Silva 

Academic Editor

PLOS ONE